# Multilayered Functional Triboelectric Polymers for Self-Powered Wearable Applications: A Review

**DOI:** 10.3390/mi14081640

**Published:** 2023-08-20

**Authors:** Minsoo P. Kim

**Affiliations:** Department of Chemical Engineering, Sunchon National University, Suncheon 57922, Republic of Korea; mspkim@scnu.ac.kr

**Keywords:** multilayered polymer, interfacial polarization, triboelectric device, multi-functionality, self-power, wearable applications

## Abstract

Multifunctional wearable devices detect electric signals responsive to various biological stimuli and monitor present body motions or conditions, necessitating flexible materials with high sensitivity and sustainable operation. Although various dielectric polymers have been utilized in self-powered wearable applications in response to multiple external stimuli, their intrinsic limitations hinder further device performance enhancement. Because triboelectric devices comprising dielectric polymers are based on triboelectrification and electrostatic induction, multilayer-stacking structures of dielectric polymers enable significant improvements in device performance owing to enhanced interfacial polarization through dissimilar permittivity and conductivity between each layer, resulting in self-powered high-performance wearable devices. Moreover, novel triboelectric polymers with unique chemical structures or nano-additives can control interfacial polarization, allowing wearable devices to respond to multiple external stimuli. This review summarizes the recent insights into multilayered functional triboelectric polymers, including their fundamental dielectric principles and diverse applications.

## 1. Introduction and Background

Wearable electric devices have been developed to monitor body motions and detect bio-signals in electric devices and can be applied in various fields, including biosensors [1,2], human–machine interfaces [3,4], and soft robotics [5]. With the demand for smart devices with high-output performance in response to external multi-stimuli, wearable technology has undergone a process of refinement and integration. The evolution has given rise to multifunctional smart wearable devices, which have found utility in Internet-of-Things (IoT) applications, including real-time healthcare monitoring [1,6,7,8,9], as well as various industrial applications, such as real-time safety management [10,11]. Therefore, the devices necessitate attributes such as flexibility, stretchability, recyclability, and the power sources for seamless operation of the ubiquitous devices. Notably, the reliance on electricity as a driving force for wearable devices has posed a challenge, particularly in terms of conventional power sources like batteries. This reliance has, in effect, imposed limitations on the advancement of self-powered wearable applications. Consequently, self-powered wearable devices have been designed that operate on energy-harvesting principles, such as electromagnetic [12], thermoelectric [13,14,15], photoelectric [16], piezoelectric [17,18], and triboelectric mechanisms [19,20,21], eliminating the need for external power sources and enabling charging–discharging cycles.

Considering the energy conversion of surrounding mechanical stimuli to electricity, triboelectric devices that operate based on the mechanism of triboelectrification and electrostatic induction have evolved (Figure 1a). The self-powered wearable systems driven by the triboelectric nanogenerators are designed for the direct detection of bio-signals responsive to different micromotions, including eye movements [22,23,24], and the device operation by a capacitor stored from the mechanical stimuli [25,26], which can be implanted anywhere on the body or cloth.

In the triboelectric process, the surface charges are generated by the contact electrification on dissimilar dielectric films during periodic contact, which are then transferred to the electrode through electrostatic induction. Consequently, a current flows through the external circuit, and alternating electric signals appear during the cyclic triboelectric process. The contact electrification is a common phenomenon that occurs at different interfaces, including the solid–solid, liquid–solid, liquid–liquid, and gas–other interfaces [27], when the triboelectric-pair materials are mechanically in contact and generate opposite surface charges. The typical charge transfer mechanisms in the contact electrification can be demonstrated by electrons [28,29], ions [30], and material transfer [31], which has been reviewed in the previous literature [32]. The electron transfer mechanism is proposed at dielectric–dielectric and metal–dielectric pair materials, which is similar to the energy bandgap theory. In the mechanism, an electron cloud overlap between dissimilar atoms under stress allows electron transition from one atom to another through a lower potential barrier, which is affected by the intrinsic properties of triboelectric materials, such as electron affinity, work function, and surface potential. The ion transfer mechanism is addressed by the transfer of mobile ions from one material to another when contacting. The ion transfer is dominated on the hydrophilic solid surface, such as an ionic polymer or nonionic polymer with absorbed ions. The formation of different degrees of ion adsorption dependent on the types of tribo-pair materials causes the charge separation, resulting in the formation of the electric double layer and, thus, potential difference, which contributes to the contact electrification. While the electrification of solid–solid and gas–solid contact is dominated by the electron transfer, that of liquid-driven contact is caused by both electron and ion transfer. Unlike the electron and ion transfer model, the material transfer mechanism can be considered in the case of polymeric materials with different mechanical properties. In 2011, it was discovered that the contact electrification between soft polymers produces nanoscopic mosaic charge patterns, as characterized by the measurement of Kelvin probe force microscopy and X-ray photoelectron spectroscopy.

Based on this triboelectric mechanism, four different types of triboelectric devices have been developed to ensure the effective conversion of external mechanical sources to electricity (Figure 1b) [32,33,34]. The first type is the contact–separation mode, which is the most common and simplest triboelectric nanogenerator. It consists of a two-electrode configuration with dissimilar dielectric layers attached to the electrodes. When the dielectric layers come into contact, opposite surface charges with the same density are generated through contact electrification. Upon separation, a potential difference is induced between the contact-pair materials, leading to the flow of the current and appearance of alternating current (AC) signals through the periodic triboelectric process. The second type is the lateral sliding mode, also with a two-electrode configuration but with a different motion compared to the contact–separation mode. It allows for free motion between the contact-pair dielectric materials with different effective contact areas. During frictional sliding, opposite charges are generated on the sliding dielectric surfaces, resulting in a potential difference between the sliding-pair layers. This process leads to the generation of AC signals through iterative sliding. The third type is the single-electrode mode, which has a unique structure compared to traditional triboelectric devices. It is composed of one output node connected to a single electrode, while the other electrode is grounded anywhere. When the triboelectric layer comes into contact with another material, surface charges are generated and transferred, inducing opposite charges on both surfaces without electron flow. Upon separation, the charges are balanced by exchanging charges with the ground. When they approach again, electron flow from the ground to the electrode occurs, producing pulse current signals. Finally, the freestanding mode is based on common friction between a moving dielectric layer and a surrounding material, similar to the single-electrode mode without the reference ground electrode. The original position is the same as that in the lateral-sliding mode. Once motion occurs, opposite charges are produced on both dielectric surfaces. Moving forward and backward induces a potential difference due to the change in overlapped area, driving the exchange of electrons between them.

The fundamental theory and applications of triboelectric devices have been extensively reviewed [35,36,37]. According to the triboelectric principles on the basis of Maxwell’s displacement current and differential surface polarization [38], the output performances of triboelectric devices are largely affected by the amounts of surface charge during the cyclic triboelectric process. Therefore, maximizing the charge density generated on the dielectric surface (contact electrification) and transferred to the electrode (electrostatic induction) is critical for improving the triboelectric performance. Approaches to increasing the surface charge density include surface engineering, including the introduction of single molecules with high electron affinity or the control of surface patterns on the triboelectric layers (Figure 2a); self-improving charge through device integration (Figure 2b); and enhancing dielectric polarization in materials (Figure 2c).

Surface functionalization, such as the injection of high-electron-affinity molecules through plasma treatment, element doping, and ion injection [40,43,44,45,46,47,48], has been attempted in order to enhance the surface charge density generated through contact electrification. Additionally, diverse patterns have been designed on the surfaces of triboelectric layers using selective etching, photolithography, and laser irradiation [39,49,50,51,52], leading to increased surface charges through larger contact areas. However, these surface modification approaches have limitations in achieving significantly higher surface charge density owing to the intrinsic electron affinity of molecules and complex patterning processes.

Another practical approach to achieving a high surface charge density is self-improvement by integrating triboelectric devices (Figure 2b). Cheng et al. and Jian et al. fabricated multiple triboelectric devices connected through a rectifier bridge [53,54]. During the periodic contact–separation process, initial charges are generated in one triboelectric device, and the charges are then accumulated and pumped up in the other triboelectric device, resulting in self-improving triboelectric devices. In addition, without any rectifier, Li et al. demonstrated a four-integrated triboelectric device [41], utilizing a self-charge excitation circuit, including diodes with directional conductivity and capacitors for charge trapping and accumulation. These multiple triboelectric designs effectively increase the charge density and output performance.

The surface charge density is proportional to the dielectric constant of the triboelectric material (Figure 2c) [55], which is affected by dielectric polarization, such as electronic, vibrational (or atomic), orientational (or dipolar), ionic, and interfacial polarization (Figure 3a) [56,57]. Researchers have explored various structural dielectric materials with high permittivity to achieve high-output performance in triboelectric devices [55,58,59]. One practical approach involves incorporating high-permittivity nanoparticles into dielectric materials in a simple manner to increase the dielectric constant, inducing interfacial polarization at the interfaces between the dielectric matrix and nanoparticles due to the mismatch in permittivity and conductivity between dissimilar materials. However, there is an upper limit to the dielectric constant improvement using this method, as previously reported [60]. Another approach involves the heterogeneous layer stacking of dielectric films with significant differences in permittivity and conductivity. Layer-structured films exhibit higher dielectric constants than single-layer films [61], surpassing the surface charge density and enhancing the triboelectric performance.

The concept of layer-structured dielectric films for enhanced polarization has been previously introduced [61,62,63], which has been employed in energy-storage applications [64,65], but its potential in self-powered triboelectric applications remains unexplored. The multilayered dielectric materials have significant potential for self-powered triboelectric applications with multifunctional and high-output performances, although there are few reports on multilayer-structured dielectric polymers. This review focused on functional multilayered dielectric materials for self-powered triboelectric applications (Figure 3b).

## 2. Multilayer-Structured Triboelectric Materials with Multifunctionality

Polymeric materials responsive to surrounding stimuli have been extensively developed for multifunctional wearable devices, necessitating flexibility, high sensitivity in response to external stimuli, and adaptive integration processes [1,66]. In addition, polymers with high electron affinity, such as fluorinated polymers and polydimethylsiloxane, Refs. [59,67,68] have been used as negative triboelectric materials for self-powered wearable devices based on the triboelectric mechanism.

Because the triboelectric performance is primarily affected by the amount of surface charge generated during the cyclic triboelectric process and their electrostatic induction to the electrode, their ability to generate and store charges in the triboelectric pair materials is crucial for enhancing output performance. Triboelectric materials have been classified based on their electron affinity (Figure 4a) [69,70], with those possessing electron-withdrawing capabilities (or Lewis acidic properties) considered negative triboelectric materials and those with electron-donating abilities (Lewis base properties) regarded as positive triboelectric materials (Figure 4b). In general, halogen- or silicone-containing dielectric materials with strong electron-withdrawing properties have been used as negative triboelectric materials. In contrast, amine-containing materials and metals with good electron-donating properties have been used as positive triboelectric materials. The cyclic contact of triboelectric pair materials with single-layer structures induces surface charge polarization, resulting from a difference in the triboelectric polarity of contact pair materials, which restricts the improvement of the triboelectric performance owing to their intrinsic properties, considering their electron donating/withdrawing ability.

To further improve triboelectric performance, researchers have developed multilayer structured films. This approach increases the dielectric constant through high-induction interfacial polarization. Interfacial polarization takes place when space charges (electrons and ions) gather at the interfaces between two dissimilar materials with significant differences in their ability to conduct an electric field [71]. Multilayer polymer dielectrics that boost their dielectric constant through interfacial polarization between distinct layers have been extensively studied [62]. This polarization in the multilayer films can be easily managed at the interface perpendicular to the electric field. It is attributed to the accumulation of the charge at these interfaces. This process significantly elevates the dielectric constant, resulting in improved surface charge density and, thus, enhanced triboelectric performance. Recent progress in these multilayer configurations, including functional polymers, has enabled their application in layer-by-layer stacking as triboelectric materials for versatile self-powered wearable devices. In the subsequent section, multilayer-structured materials with enhanced triboelectric performance have been introduced.

### 2.1. Polymer-Based Multilayer Structures for Enhanced Triboelectric Performances

As mentioned above, the layer-by-layer stacking of dielectric films with significant differences in permittivity and conductivity effectively enhances the dielectric constant through increased interfacial polarization, leading to high surface charge density and improved triboelectric performance. In this section, we review dielectric-polymer-based multilayer structures to enhance triboelectric performance through improved interfacial polarization.

Initially, layer-structured dielectric films were introduced into triboelectric devices, serving as charge trapping/storing abilities in intermediate films (Figure 5a–c). Shao et al. introduced an SiO_2_ layer between two dielectric layers (parylene C and polyimide) in a metal-to-dielectric triboelectric device [72]. Adding an intermediate layer resulted in a higher output performance of the three-layered dielectric film than the two-layered one, attributed to the effective charge polarization in the layer-structured film. Additionally, Kim et al. [73] and Feng et al. [74] demonstrated the effect of an intermediate film positioned between the top dielectric film and the bottom electrode (Figure 5b,c). In the bilayer film, an intermediate film with a dielectric property lower than that of the top layer facilitates charge trapping or storage in the dielectric film, leading to an increased charge density. In the contact–separation mode triboelectric devices, the contact electrification occurs at the solid–solid interfaces, producing opposite surface charges on the triboelectric layers. The addition of intermediate layers highly induces the polarization owing to different permittivity and conductivity between the dielectric and intermediate layers, resulting in the charge accumulation at the dielectric interfaces. After all, the surface charges increased depending on the kinds of intermediate materials, leading to greatly increased triboelectric performances. The fundamental mechanism of performance improvement can be demonstrated by electron- and ion-transfer caused by space charge polarization. The intermediate dielectrics for triboelectric devices have been reviewed in the literature [58].

Recently, our group systematically demonstrated the effect of bilayer films comprising polymers with different fluorine units and soft/hard insulating layers on triboelectric performance (Figure 5d,e) [75,76]. In these studies of using triboelectric devices with contact–separation mode, fluorinated polymers with three fluorine units in the side chain (poly(2,2,2-trifuoroethyl methacrylate), PTF) were coated on insulating substrates with a lower dielectric constant, increasing the dielectric constant. This enhancement was attributed to the improved interfacial polarization at the interface between the semi-crystalline PTF and the insulating layer. Consequently, the PTF-coated bilayer film exhibited significantly higher triboelectric performance compared to other fluorinated polymer films (Figure 5d) [75].

More specifically, we investigated the factors influencing the interfacial polarization for the high triboelectric performance of bilayer films (Figure 5e) [76]. In addition to the mismatched dielectric and electric properties of each layered film, the film thickness plays a crucial role in improving interfacial polarization. While the thicker dielectric film disturbs the polarization, the thinner dielectric film causes the leakage current through the flow of polarized charges to the electrode. After all, it results in the suppression of the dielectric constant and the resultant triboelectric performance. Therefore, an optimal film thickness is required to effectively induce polarization. Figure 6b shows a soft bilayer film comprising PTF-coated polydimethylsiloxane (PTF-PDMS), fabricated as a negative triboelectric material that exhibits a highly enhanced output performance in a triboelectric device when the PDMS thickness is ~33 μm. Additionally, we fabricated a poly (L-lysine)-coated PDMS bilayer film (PLL-PDMS) as a positive triboelectric material, resulting in improved triboelectric performance when the PDMS thickness was ~10 μm. Subsequently, the triboelectric device comprising soft bilayer films as negative and positive triboelectric materials exhibited the best triboelectric performance compared to PDMS-based triboelectric devices [76].

These results suggest that heterogeneous dielectric multilayer films can be a robust design for enhancing the triboelectric performance of flexible or wearable devices. However, it is still challenging to develop self-powered functional wearable devices consisting of multilayer dielectric films based on the triboelectric mechanism.

### 2.2. Multilayered Functional Composites for Enhanced Triboelectric Performances

In addition to polymer-based multilayer films, layer-stacked films comprising functional polymer composites with high-permittivity additives, such as barium titanate (BTO) nanoparticles, carbon nanotubes (CNT), carbon black, and metallic nanowires, offer increased interfacial polarization owing to the significant contrast in permittivity and conductivity. Therefore, multilayer films, including functional polymer composites, allow for the significant enhancement of triboelectric performance and the development of novel triboelectric applications.

Li et al. demonstrated multilayered triboelectric nanogenerators comprising electrospun fibers as positive and negative triboelectric layers (Figure 6a) [77]. It should be noted that the charge transport and storage layers were layer-stacked in the negative triboelectric film, resulting in higher triboelectric performance compared to the nanogenerators without the conducting or insulating layers. The basic mechanism can be demonstrated by a similar mechanism coupled with the space charge polarization, as mentioned in Figure 5a–c. This increased polarization through the charges accumulated in the multilayered negative triboelectric films, with significant differences in permittivity and conductivity, contributed to high-performance triboelectric devices. Recently, our group fabricated a multilayer-structured dielectric film by alternating the stacking of poly(vinylidene fluoride–trifluoroethylene) (PVDF-TrFE) with low permittivity and BTO nanoparticles with high permittivity (Figure 6b) [78]. The PVDF-TrFE/BTO multilayered film displayed superior dielectric properties compared to the single PVDF-TrFE and PVDF-TrFE/BTO composites, resulting in increased triboelectric performance, attributed to enhanced polarization through the reduced leakage current and efficient stress concentration effects of the alternating soft (PVDF-TrFE)/hard (BTO) layers. Subsequently, in this study, four-layered PVDF-TrFE/BTO film with a thickness of ~50 μm demonstrated the best triboelectric performance.

Self-powered wearable devices with self-healing abilities have been developed, inspired by human skin with repetitive damage-healing capabilities. However, self-healing polymeric materials inherently possess poor triboelectric and dielectric properties because of their fundamental chemical and physical properties arising from their intrinsic chain composition [69,79], leading to the suppression of triboelectric performance. Researchers have explored modifications in the chemical composition, including high electron affinity, such as fluorine units [80,81], and the addition of high-permittivity moieties, such as carbon nanotubes (CNTs) and ionic liquids, to enhance the output performance of self-healing materials [81,82,83]. Triboelectric devices comprising various self-healing materials have been reviewed in the literature [84,85].

**Figure 6 micromachines-14-01640-f006:**
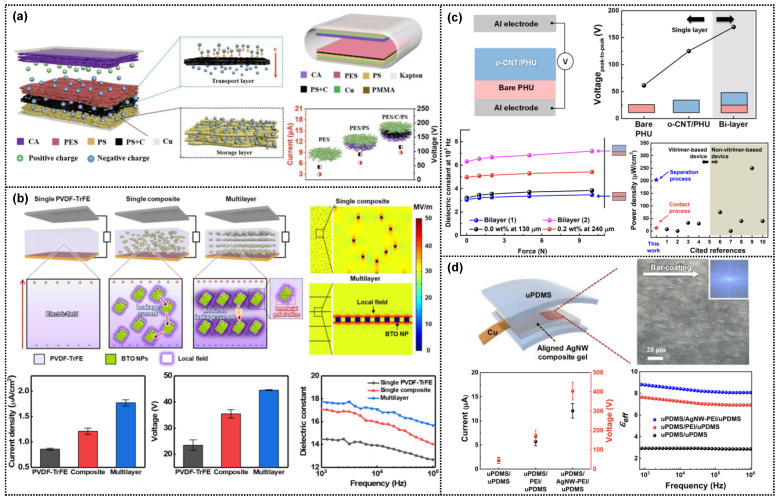
(**a**) Triboelectric nanogenerator based on multilayered fibers prepared by electrospinning cellulose acetate (CA), polyethersulfone (PES), polystyrene (PS), and PS-carbon black (PS-C) solutions, respectively (reproduced with permission from [77]. Copyright© 2018 Elsevier). (**b**) Triboelectric nanogenerator based on multilayer-structured dielectric film, including BTO nanocomposite layers (reproduced with permission from [78]. Copyright© 2020 American Chemical Society). (**c**) Self-healable triboelectric device composed of bilayered polymer composite film (reproduced with permission from [82]. Copyright© 2020 American Chemical Society). (**d**) Triboelectric devices based on layer-structured films, including AgNW–PEI composite gel (reproduced with permission from [24]. Copyright© 2022 Elsevier).

Recently, we developed self-healing triboelectric devices with significantly enhanced output performance, designed by contact–separation mode. Through the self-healing mechanism of dynamic covalent bonding, we synthesized vitrimeric poly(hindered-urea) (PHU) with dimethylsiloxane in the main chain as a negative triboelectric material using a reversible crosslinking network of dynamic hindered urea units with a bulky t-butyl group [82]. Adding oxidized CNTs (o-CNTs) to the PHU network increased the dielectric constant owing to the induced interfacial polarization between the o-CNTs and PHU. Additionally, the self-healing bilayer film comprising bare PHU and the PHU composites with o-CNTs consistently induced stronger interfacial polarization because of the significant difference in permittivity and conductivity, further enhancing the triboelectric performance (Figure 6c). Using another approach, we fabricated self-healing layer-structured films comprising self-healing PDMS elastomers and silver nanowire (AgNW) composite gels, designed using a single-electrode mode (Figure 6d) [24]. The self-healing polymer was prepared using the urea bonding reaction of amine-functionalized PDMS with isophorone diisocyanate. The triple-layered film sandwiched polyethyleneimine (PEI) gel between the PDMS layers induced interfacial polarization because of the difference in conductivity between PEI and PDMS, improving the triboelectric performance compared with a bilayer film consisting of PDMSs. Furthermore, adding the AgNW composite gel induced stronger interfacial polarization coupled with the polarization effect between the AgNWs and PEI gel, enhancing the triboelectric performance.

## 3. Self-Powered Functional Triboelectric Applications by Using Multilayer Dielectric Films

Multilayer structured dielectric films have been widely applied in energy-related [64] and bio-applications [86] because of the increased dielectric constant through greatly induced interfacial polarization, as explained in Section 2. In triboelectric applications, interfacial polarization-induced dielectric films have been examined, primarily focusing on the enhanced output performance rather than exploring the multi-functionality of self-powered wearable devices based on the triboelectric mechanism. In this section, we briefly introduce self-powered functional triboelectric devices comprising multilayer-structured dielectric films (Figure 7).

Among the factors affecting the interfacial polarization in multilayer-structured films, the surrounding temperature can effectively induce interfacial polarization attributed to the temperature-dependent change in capacitance through dipole rearrangement in the bilayer films. When the temperature increases, the dipoles become more activated, resulting in a more significant difference in the conductivity of each layer, leading to an increase in the dielectric constant through improved interfacial polarization [76]. Consequently, the triboelectric performance increased with the surrounding temperature. Based on this temperature dependence, we developed a temperature-responsive triboelectric sensor comprising soft PTF-PDMS and PLL-PDMS bilayer films as the negative and positive triboelectric materials, respectively (Figure 7a). In the contact with each bilayer film, the surface charges are generated at the solid–solid interfaces and then further elevated when heated, resulting in greatly induced interfacial polarization. Thereafter, the dielectric constant gradually increases, leading to great enhancement of triboelectric performance. After all, the triboelectric device consisting of elastic bilayer films exhibited superior temperature sensitivity compared to other triboelectric sensors [76].

Furthermore, the bilayer films were utilized for a triboelectric device, producing direct-current (DC) signals without any electronic support (Figure 7b). Triboelectric devices usually exhibit alternating-current signals during the cyclic triboelectric process without rectifying. However, in this study, the vibration-induced triboelectric nanogenerator consisting of soft bilayer films operated without a rectifier bridge, although the DC signals appeared at high frequencies, continuously illuminating an LED light. This phenomenon was attributed to charge accumulation in the bilayer films due to the slower charge relaxation time during vibration, leading to sustainable polarization through repetitive charge regeneration and accumulation in the dielectric film.

Functional dielectric nanomaterials with high permittivity enable wearable devices with multifunctionality and improved output performance. As shown in Figure 6b, the multilayered films comprising alternating PVDF-TrFE and BTO layers effectively induced interfacial polarization owing to the difference in permittivity and conductivity, leading to high triboelectric performance. Flexible multilayered triboelectric sensors with high sensitivity benefit self-powered wearable healthcare devices for detecting bio-signals, such as carotid and radial artery pulse pressures, human breath, and acoustic waves (Figure 7c) [78]. A triboelectric sensor comprising a multilayered PVDFTrFE/BTO film sputtered with a Pt electrode and an Al-coated PET film was utilized to detect radial artery pressure on the wrist, carotid artery pressure on the neck, and human breath pressure on the nostrils. Moreover, the multilayered triboelectric sensor responds to high-frequency stimuli ranging from 100 Hz to 8 kHz, making it a more flexible acoustic sensor. The multilayered sensors displayed the maximum output performance at a resonance frequency of 400 Hz. Referring to the frequency-dependent device performance, the triboelectric sensor detected a sound source “triboelectric sensor” based on the sound waveform and short-time Fourier transform signals. Multilayered triboelectric sensors have the potential to be used in various applications, such as self-powered microphones and voice security systems.

In the bioelectric signals detected by wearable sensors, human eye motions in various directions change the electric signals originating from the potential difference between the retina and cornea of the eye, which constitutes an electrooculogram [87,88,89]. Electrooculography (EOG) signals in response to eye movements have been successfully employed in human–computer interfaces [22,90]. Triboelectric electrooculogram (EOG) sensors based on a sliding single-electrode mode offer several advantages over other sensor types. They are relatively simple to design, cost-effective, and noninvasive. Anisotropic dielectric nanowires, unlike isotropic nanoparticles, exhibit differentiated electric signals depending on the direction of the applied external field [24]. As shown in Figure 6d, the triple-layered triboelectric sensor with an aligned AgNW-composite gel displayed distinct electric signals through differentiated polarization dependent on the direction of eye motion, making it ideal for self-powered wearable EOG sensors. The fabricated EOG sensor detected eight-directional eye motions (from the center to the upper left, left, lower left, down, right, upper right, up, and lower right), with each triboelectric signal corresponding to the direction of eye motion caused by the intrinsic dielectric properties of the eye with positive dipoles (cornea) and negative dipoles (retina). The working mechanism could be demonstrated by electron- and ion-transfer through space charge polarization dependent upon the tactile directions. In addition to the direction-dependent triboelectric signals, the EOG sensor (single-electrode mode triboelectric sensor) exhibited different signal polarities and response times depending on the direction of motion enabling the precise detection of eye movements without external power sources. It was attributed to the distinctive charge transfer time after the contact electrification. Hence, a multilayer-structured triboelectric device comprising a self-healable PDMS layer and aligned AgNW composite gel represents a significant advancement over previously developed EOG sensors [22,88,89,91,92].

For more practical triboelectric applications, researchers manufactured multilayered functional films using a simple layer-by-layer (LbL) technique [93]. Menge et al. created multilayered dielectric films through an electrostatic-assisted assembly of poly(ethylene oxide) and poly(acrylic acid) ([PEO/PAA]_n_) using the LbL method. This resulted in ultrathin, stretchable, and freestanding triboelectric materials. The operational principle of the multilayer-based triboelectric device was demonstrated to involve electron transfer with the contact–separation mode. The multilayered film, optimized for its thickness and number of layers, exhibited a strong ability to donate electrons without interference from electrostatic induction charges, leading to an increase in capacitance. Specifically, the [PEO/PAA]_20_ configuration demonstrated enhanced surface charge, thereby improving triboelectric performance. Leveraging the exceptional geometric, mechanical, and optical properties of these multilayer films, freestanding triboelectric devices with a single-electrode mode were designed that could adapt to various shapes, including kirigami-type films. These versatile and customizable devices could be affixed to a variety of surfaces to harness energy from mechanical movements, thus enabling the development of wearable energy-harvesting and storage applications. Additionally, due to their responsiveness to environmental stimuli, self-powered skin-like sensors were invented, which could be attached or implanted on diverse substrates. Despite these advancements, challenges persist in the development of layer-structured materials, including flexible functional nanocomposites, for self-powered multifunctional wearable applications.

## 4. Conclusions and Perspectives

Multilayered dielectric materials offer significant potential for multifunctional self-powered wearable applications due to their flexibility, adjustable dimensions, and controllable polarization. These materials play a crucial role in advancing IoT technology, which demands miniaturization and multifunctionality. The triboelectric effect, a common phenomenon in daily life, holds promise for harvesting energy in self-powered wearable devices, primarily by influencing surface polarization. Dielectric films with notable disparities in permittivity and conductivity between distinct layers effectively induce interfacial polarization. This, in turn, leads to a high dielectric constant and augmented surface charge density, thereby enhancing triboelectric performance. Moreover, functional layer-structured dielectric films, comprising functional polymers and high permittivity nano-additives, facilitate the creation of multifunctional self-powered wearable devices. These advancements pave the way for next-generation smart wearable applications with innovative functionalities, such as simultaneous haptic feedback, heightened sensitivity to bio-signals, sight-through haptics, and integrated functions.

Until now, research has primarily concentrated on the advancement of tribo-pair dielectric materials, typically involving negative triboelectric materials. Recognizing the potential of multilayered dielectric materials to amplify triboelectric performance through the interplay between surface charge density and dielectric constant, the innovation lies in combining triboelectric materials with diverse polarization mechanisms. This integration paves the way for the creation of multifunctional self-powered wearable devices with high sensitivity, which can be achieved as follows:For effective multifunctional IoT applications, the development of self-powered wearable devices exhibiting high-output performance is crucial. Multilayer structured materials offer significant potential due to their remarkable tunable polarization. However, there is little research to systematically investigate layer-structured films that promote highly induced interfacial polarization, such as exploring suitable dielectric materials, optimizing layer configurations, and establishing comprehensive models for such layered materials. Additionally, despite the availability of numerous functional additives, such as nanoparticles and nanowires, the utilization of functional layer-structured films designed through optimal layer-stacking approaches has been largely unexplored. This untapped potential holds promise for driving next-generation IoT technology forward.Alongside achieving optimal layer structures, the simplicity of the fabrication process for multilayered films holds significance. The majority of preparation processes involve techniques such as spin-coating, bar-coating, spray-coating, and successive deposition, demanding multiple steps for layer stacking. For the large production of multilayer structured films, it is vital to develop streamlined fabrication processes.Much of the research has centered around negative triboelectric materials. However, since triboelectric performance stems from the contact electrification between positive and negative triboelectric layers, the significance of positive triboelectric materials cannot be overlooked in the quest to improve output performance. The utilization of polarization-induced triboelectric pair materials with multilayer structures holds the potential to advance the field of triboelectric devices, resulting in notably enhanced output performance. This advancement is beneficial for practical applications that demand high output power, such as smart wearable devices and portable IoT devices.The practical applications for multilayer-based triboelectric devices should be further developed. While much of the research has focused on enhancing output performance, the immense potential of multilayer structured materials, as highlighted earlier, suggests the possibility of creating flexible smart wearable devices with both multifunctionality and high performance. Such advancements will align with the ultimate goals of next-generation IoT applications.

## Figures and Tables

**Figure 1 micromachines-14-01640-f001:**
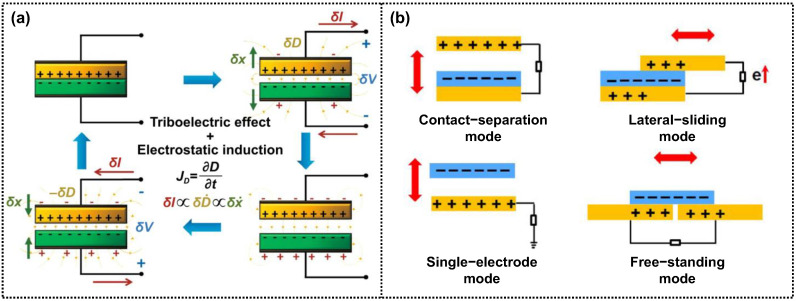
(**a**) Schematic representation of the working cycle of a contact–separation TENG based on the combined phenomena of triboelectric effect and electrostatic induction. In the figure, *δx, δV, δI,* and *δD* represent a change in the distance between the triboelectric layers, a change in voltage and current in an external circuit, and a change in electric displacement field, respectively. The plus and minus signs represent the electrostatic triboelectric charges on the triboelectric surfaces (in black) and the free charges on the electrodes (in red), respectively [21]. Copyright© 2021 Springer Nature. (**b**) Different working modes of triboelectric devices that have been developed so far.

**Figure 2 micromachines-14-01640-f002:**
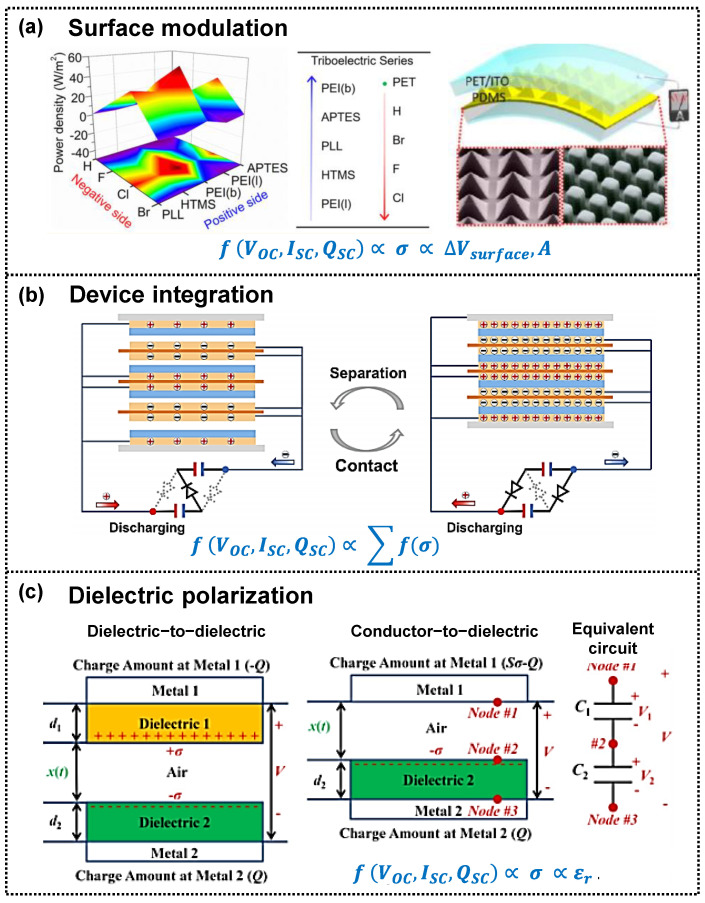
Representative methods to improve the surface charge density by (**a**) surface modulation (reproduced with permission from [39,40]. Copyright© 2012, 2017, American Chemical Society), (**b**) device integration (reproduced with permission from [41]. Copyright© 2023 Elsevier), and (**c**) dielectric polarization (reproduced with permission from [42]. Copyright© 2015 Elsevier), which can enhance the triboelectric performance. In this figure, *V_OC_, I_SC_, Q_SC_, σ, ΔV_surface_*, and *A* indicate open-circuit voltage, short-circuit current, short-circuit transferred charge, surface charge, surface potential, and contact area, respectively.

**Figure 3 micromachines-14-01640-f003:**
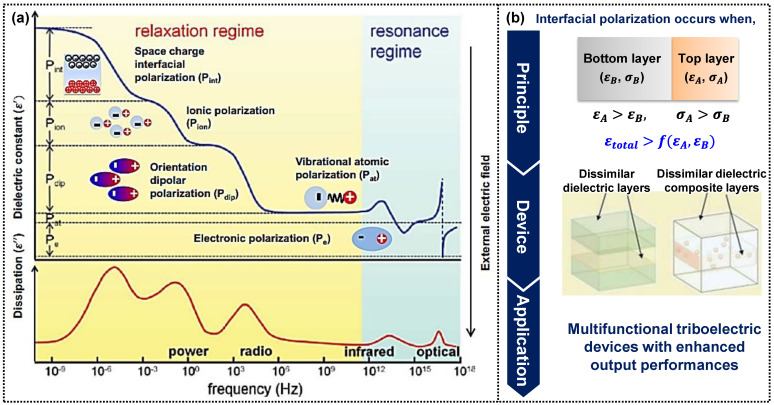
(**a**) Real (ε′) and imaginary part (ε″) of the dielectric constant as a function of frequency in a polymer having interfacial, orientational, ionic, and electronic polarization mechanisms (reproduced with permission from [56]. Copyright 2012 American Chemical Society). (**b**) The focus of this review is for multilayered functional triboelectric polymers for self-powered wearable devices with high-output performances through highly induced interfacial polarization.

**Figure 4 micromachines-14-01640-f004:**
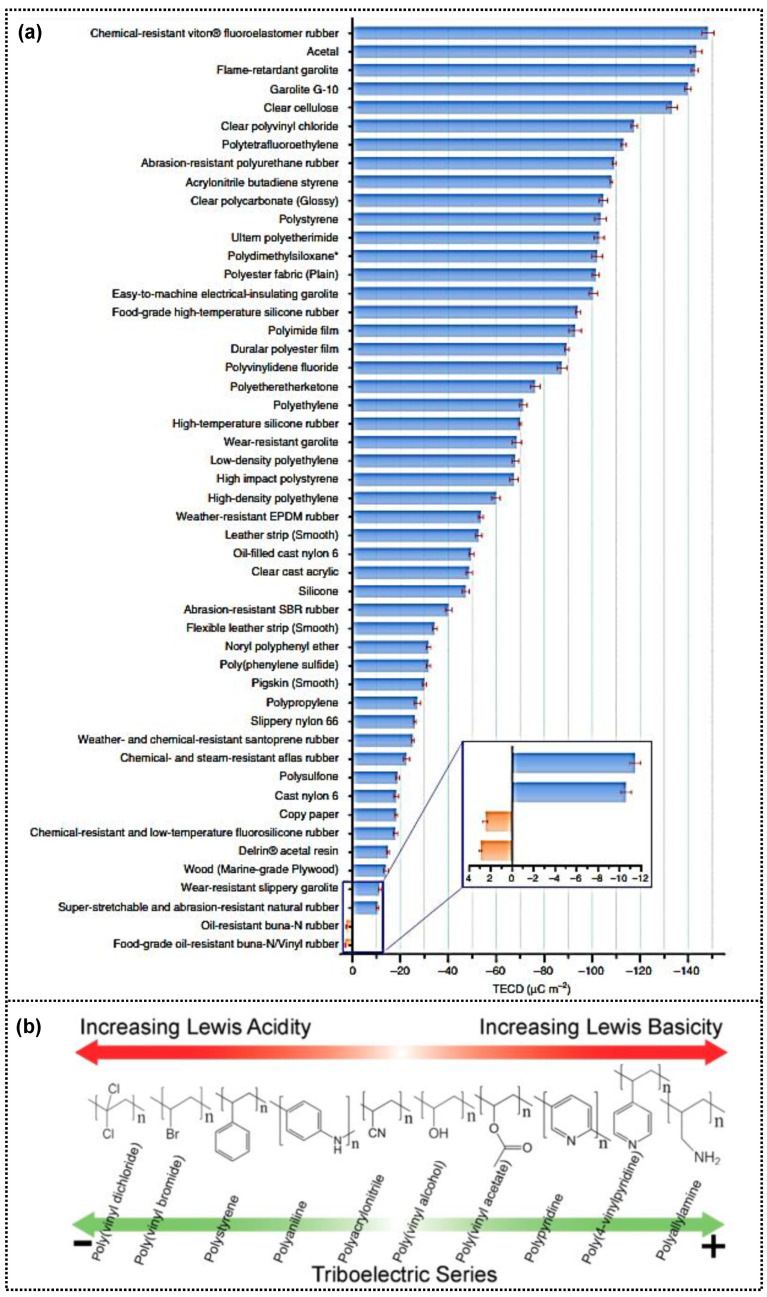
(**a**) The quantified triboelectric series (reproduced with permission from [69]. Copyright© 2019 Springer Nature). (**b**) The charge density of tribo-pair polymers after contact process and the correlation between the ordering of triboelectric materials and Lewis basicity/acidity (reproduced with permission from [70]. Copyright© 2019 American Chemical Society).

**Figure 5 micromachines-14-01640-f005:**
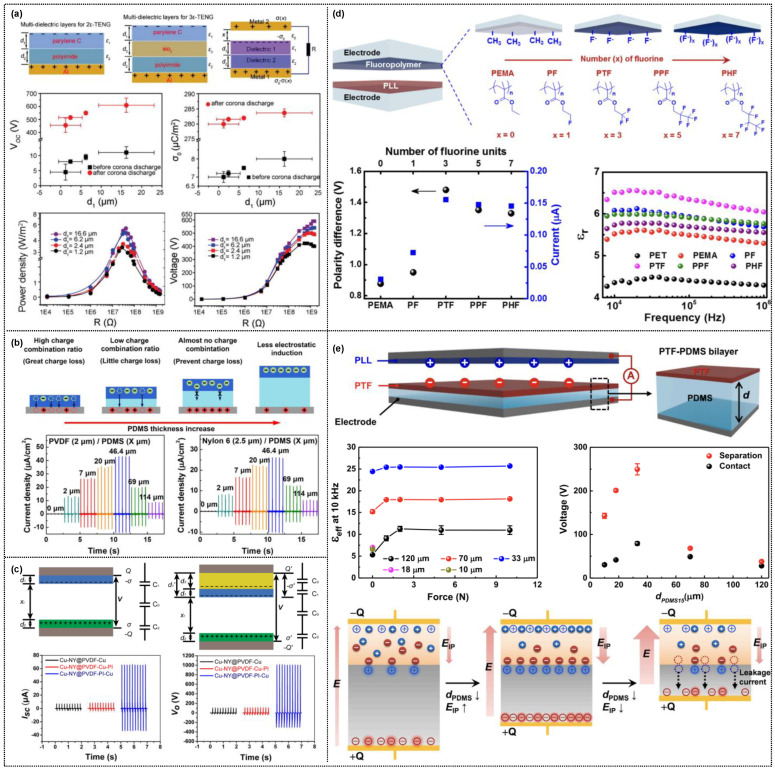
(**a**) Schematic diagram of layer-structured dielectric films for bi- and triple-layered triboelectric devices composed of PC/PI layers and parylene C/SiO_2_ PI layers, respectively, displaying the triboelectric performance before and after corona discharge (reproduced with permission from [72]. Copyright© 2017 Royal Society of Chemistry). (**b**) Triboelectric nanogenerator consisting of PVDF/PDMS double layer and Nylon 6/PDMS double layer with various PDMS interlayer thicknesses (reproduced with permission from [73]. Copyright© 2018 Elsevier). (**c**) Triboelectric nanogenerator of layer-stacked films without and with PI as a transition layer for charge storage (reproduced with permission from [74]. Copyright© 2017 Elsevier). Bilayer-structured triboelectric nanogenerators based on fluorinated polymers with different kinds of fluorine units coated on (**d**) hard PET-insulating dielectric film (reproduced with permission from [75]. Copyright© 2018 Elsevier) and (**e**) soft PDMS-insulating dielectric film (reproduced with permission from [76]. Copyright© 2021 Elsevier). To effectively induce the interfacial polarization for high-performance triboelectric nanogenerators, (**d**,**e**) demonstrate the effect of semicrystalline fluorinated polymers and the dielectric insulating film between fluorinated polymer and electrode, respectively.

**Figure 7 micromachines-14-01640-f007:**
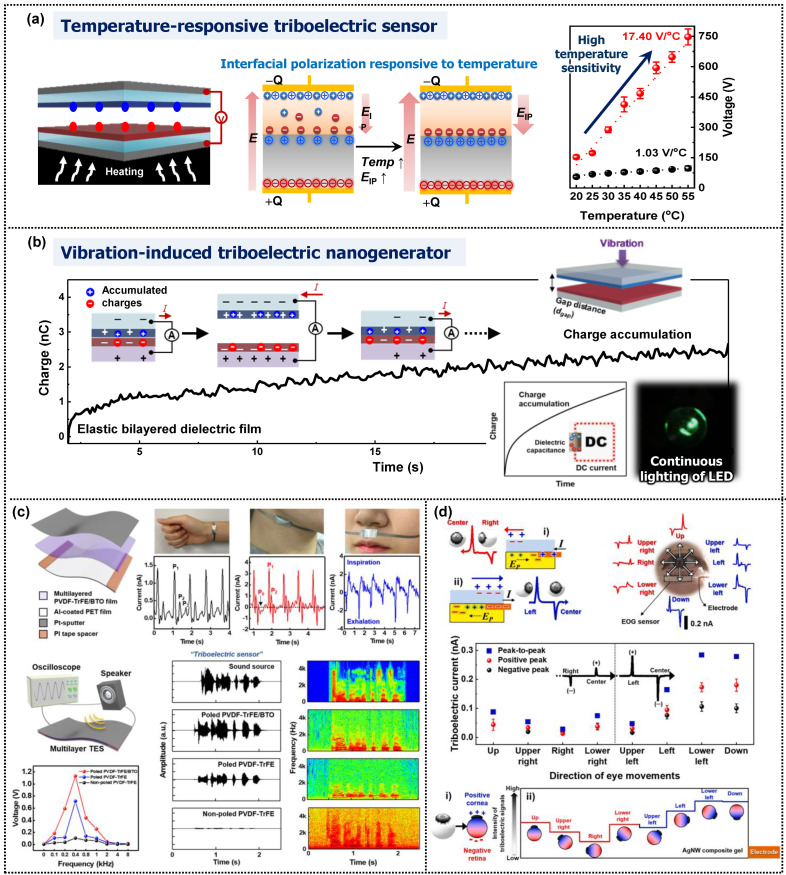
Triboelectric devices consisting of bilayer-structured films with greatly induced interfacial polarization; (**a**) a temperature-responsive triboelectric sensor with the best temperature sensitivity and (**b**) a vibration-induced triboelectric nanogenerator displaying DC current signals without any electronic support (reproduced with permission from [76]. Copyright© 2021 Elsevier). (**c**) Multilayered PVDF-TrFE/BTO triboelectric device and the self-powered wearable sensors of multilayered device (reproduced with permission from [78]. Copyright© 2020 American Chemical Society). (**d**) Schematic illustration of the operational mechanism of EOG sensor consisting of layer-structured films and the triboelectric signals in response to eye motions (reproduced with permission from [24]. Copyright© 2022 Elsevier).

## Data Availability

All data are presented in this study.

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
