# Peer review of "Multilayered Functional Triboelectric Polymers for Self-Powered Wearable Applications: A Review"

_micromachines, 2023, doi:10.3390/mi14081640_

Round 1
Reviewer 1 Report
The manuscript discusses multilayered functional triboelectric polymers for self-powered wearable applications. It highlights the need for flexible materials that can obtain electric signals and monitor body motions in wearable devices. The authors are required to provide relevant feedback on the following comments.
- The logical architecture of this literature is basically clear, but there are some areas that can be further improved. Firstly, the manuscript should provide a more comprehensive introduction with more references to the current applications and requirements of wearable devices to facilitate readers' understanding of the research's context and significance. Secondly, a more detailed explanation of its influence on the triboelectric properties can be provided when considering the introduction of multilayer structured dielectric films. Thirdly, exact examples and cases can be presented to demonstrate the potential applications of multi-layer functional triboelectric polymers, thereby supporting their feasibility and potential in practical scenarios. Additionally, to further improve the logical structure, it could benefit from a more explicit organization of the different sections or subheadings. This would help readers navigate the review more easily and understand the progression of ideas.
- The question about that how do multilayered structures of dielectric polymers effectively induce interfacial polarization and enhance surface charge density can be discussed in a more detail way?
- The question about that are there any potential drawbacks or limitations that need to be addressed regarding the use of multilayered functional triboelectric polymers in self-powered wearable applications should be discussed.
- Could you elaborate on the potential manufacturing challenges or considerations in multilayered dielectric films with flexibility, tunable dimensions, and controlled polarization?
- Detailed literature review about contact electrification and self-powered system in the introduction part can improve the quality of manuscript, such as Adv. Funct. Mater. 33, 2213410, 2023; Sci. Adv. 3, e1700694, 2017; J. Phys. Chem. Lett. 13, 6721, 2022; Micromachines, 12, 352, 2021, and so on.
The manuscript is well organized and written.
Author Response
We greatly appreciate the valuable comments from reviewers. As attached a file below, where we provide point-by-point responses to their comments, we have made necessary changes to the manuscript, addressing all the reviewers’ comments. Please find enclosed the revised manuscript entitled “Multilayered functional triboelectric polymers for self-powered wearable applications: A review”. The changes are yellow-highlighted. Further, all suggestions from you and the editorial office have been addressed.
We hope that the revised manuscript meets with your approval as well as the reviewers. We look forward to hearing positive feedbacks and please let me know if you have any questions regarding manuscript.

Reviewer 2 Report
This review summarized the recent advancement of multilayered functional polymer-based triboelectric nanogenerator device on self-powered wearable application. As a mini review paper, this paper discussed the topic from three aspects: polymer-based multilayer structures, multilayered functional composites, and self-powered functional triboelectric applications. The paper organization is relatively clear and brief, and can be well readable. However, some concerns should be addressed before further consideration. The specific comments are listed below for your reference.
1. In the paper organization, the author may show more discussions on multilayered polymer structures.
2. I suggest the authors would better focus on the working principles of different TENGs. When explaining the performance improvement, the basic mechanism should be well interpreted and compared.
3. When reviewing the applications of TENGs in wearable electronics, more typical demonstrations should be implanted, such as, DOI:10.1002/adfm.202204803.
Author Response

(The authors gave the same response as above.)

Reviewer 3 Report
An analysis of a topical issue is presented. The state of the art is adequate and oriented in the line of interest of the study.
Some figures, it is advisable to improve the quality (Fig. 1a, 2b, 2c, 4a, 5, 6). Where texts are not clear, some are very loaded. Perhaps consider better organization or present it as supplementary material. Fig.7 is presented in adequate quality, but not other figures.
An interesting topic is presented, some changes especially in the quality of the figures and discussion. They may be needed.
Author Response

(The authors gave the same response as above.)

Round 2
Reviewer 1 Report
The authors have addressed all the issues from the reviewers. Therefore, I recommend it to be accepted for publication in this form.